# Meridianins Inhibit GSK3β In Vivo and Improve Behavioral Alterations Induced by Chronic Stress

**DOI:** 10.3390/md20100648

**Published:** 2022-10-19

**Authors:** Anna Sancho-Balsells, Esther García-García, Francesca Flotta, Wanqi Chen, Jordi Alberch, Manuel J. Rodríguez, Conxita Avila, Albert Giralt

**Affiliations:** 1Departament de Biomedicina, Facultat de Medicina, Institut de Neurociències (UBneuro), University of Barcelona, 08036 Barcelona, Spain; 2Institut d’Investigacions Biomèdiques August Pi i Sunyer (IDIBAPS), 08036 Barcelona, Spain; 3Centro de Investigación Biomédica en Red Sobre Enfermedades Neurodegenerativas (CIBERNED), 08036 Barcelona, Spain; 4Production and Validation Center of Advanced Therapies (Creatio), Faculty of Medicine and Health Science, University of Barcelona, 08036 Barcelona, Spain; 5Department of Evolutionary Biology, Ecology and Environmental Sciences, Faculty of Biology and Biodiversity Research Institute (IRBio), University of Barcelona, 08028 Catalonia, Spain

**Keywords:** GSK3β, PKA, PKC, Akt, GluR1, memory, synaptic activity

## Abstract

Major depression disorder (MDD) is a severe mental alteration with a multifactorial origin, and chronic stress is one of the most relevant environmental risk factors associated with MDD. Although there exist some therapeutical options, 30% of patients are still resistant to any type of treatment. GSK3β inhibitors are considered very promising therapeutic tools to counteract stress-related affectations. However, they are often associated with excessive off-target effects and undesired secondary alterations. Meridianins are alkaloids with an indole framework linked to an aminopyrimidine ring from Antarctic marine ascidians. Meridianins could overcome several of the aforementioned limitations since we previously demonstrated that they can inhibit GSK3β activity without the associated neurotoxic or off-target effects in rodents. Here, we show that meridianins delivered into the lateral ventricle inhibited GSK3β in several brain regions involved with stress-related symptoms. We also observed changes in major signaling pathways in the prefrontal cortex (Akt and PKA) and hippocampus (PKC and GluR1). Moreover, meridianins increased synaptic activity, specifically in the CA1 but not in the CA3 or other hippocampal subfields. Finally, we chronically treated the mice subjected to an unpredictable mild chronic stress (CUMS) paradigm with meridianins. Our results showed improvements produced by meridianins in behavioral alterations provoked by CUMS. In conclusion, meridianins could be of therapeutic interest to patients with stress-related disorders such as MDD.

## 1. Introduction

Major depression disorder (MDD) is a debilitating mental illness that affects millions of people worldwide [1]. MDD is characterized by low mood, a feeling of worthlessness, agitation, and diminished interest in activities, among others [1]. Additionally, MDD is very comorbid with other disorders such as anxiety [2]. One of the most important environmental risk factors associated with MDD is chronic stress [3,4]. Some brain regions such as the hippocampus, the nucleus accumbens (Nacc), the medial prefrontal cortex, and the amygdala interpret what is stressful and regulate an appropriate response that, when failing, may lead to major depression [5,6]. Despite years of research, current medications are still ineffective in 30% of patients and are often associated with significant side effects [7]. Therefore, it is an emergency to design more effective therapeutic approaches.

One of the molecules that have been recently associated with stress-related pathologies is the glycogen synthase kinase 3 B (GSK3β) [8,9]. GSK3β is a serine (Ser)/threonine (Thr) kinase highly enriched in the brain. GSK3β is constitutively active, and its regulation is mainly mediated by phosphorylation and dephosphorylation processes [10]. Thus, the phosphorylation of the N-terminal Ser9 inhibits GSK3β activity [10]. Akt is the major regulator of GSK3β, as it exerts inhibitory phosphorylation on Ser9 in the amino-terminal part of the protein. Additionally, other kinases such as PKA and PKC also regulate GSK3β activity [11].

Several studies showed increased activity of GSK3β in mice subjected to different stress protocols [12,13,14]. In line with this, some groups have found that blocking GSK3β along the stress protocol induces recovery in some of the deficits observed by the stress itself [12,14]. Furthermore, increased activity of this kinase has been found in depressed human post-mortem samples and peripheral tissues such as peripheral blood mononuclear cells (PBMCs) [15,16,17]. These human studies suggest that GSK3β activity correlates with the severity of maniac and depressive symptoms [15,16,17].

With all this evidence, GSK3β has emerged as a potential therapeutic target for mood disorders. For years, lithium has been the most common inhibitor used to modulate GSK3β [18,19]. However, the use of lithium raises some concerns. First, the therapeutic dose of lithium is lower than the IC50 of lithium for inhibition of GSK3 [20]. Secondly, lithium has other intracellular actions [21] and can induce some side effects such as a higher risk of developing dementia [22]. Third, the use of other GSK3β inhibitors has been associated with excessive undesired and secondary effects [23,24], causing the research for kinase modulators to reach an impasse. Therefore, the search for specific GSK3β inhibitors with no off-target effects is mandatory.

Marine natural products (MNPs) are a still understudied source of potentially bioactive compounds [25,26]. Among them, Antarctic MNPs are even more unknown regarding their biological function and their putative pharmacological role as drugs, mainly due to the difficulties in their collection, identification, and further laboratory synthesis [27,28,29,30,31]. Meridianins are a family of indole alkaloids consisting of an indole framework linked to an aminopyrimidine ring, described from marine benthic organisms from Antarctica, particularly from *Aplidium* ascidians [32,33,34,35,36,37]. In general, the species of the tunicate *Aplidium* Savigny, 1816 are the source of abundant nitrogen-containing metabolites belonging to unprecedented structural families of MNPs [30,38]. Meridianins have been isolated from different specimens of the tunicate genera *Aplidium* and *Synoicum* [36,37]; however, it is still unclear whether tunicates are the true producers of the molecules or whether the associated microbes may play a role in their synthesis and chemical ecology [28,39]. The potential of meridianins as bioactive drugs has been reported previously [33,40,41,42,43,44,45,46,47,48,49,50,51,52,53]. Recently, we showed that meridianins can act as the ATP-competitive or non-ATP-competitive inhibitors of GSK3β in vitro without altering neuronal survival [54]. Moreover, we also demonstrated that in vivo, meridianin administration can improve cognitive deficits and rescue the loss of spine density in the 5xFAD mouse model of Alzheimer’s disease [31].

In the present study, we show how the intraventricular administration of meridianins is capable of modulating GSK3β activity in several brain regions involved with stress and major depression. We also describe how meridianins regulate synaptic activity in hippocampal slices, and we identified some of the potential up- and downstream molecular targets affected. Finally, we show that in vivo GSK3β inhibition could have beneficial effects on some of the cognitive and depressive-like deficits provoked by chronic unpredictable mild stress (CUMS), which is a widely used model of induced depression [55,56].

## 2. Results

### 2.1. In Vivo Inhibition of GSK3β by Meridianins

GSK3β hyperactivity has been described in different brain regions in stress-related pathologies [13,14]. Here, we first wanted to analyze GSK3β activity in chronically stressed mice. To this end, we subjected the mice to the chronic unpredictable mild stress protocol (CUMS) for 28 days (Figure 1A). Then, 24 h after the last stressor, the mice were sacrificed to evaluate GSK3β activity. We analyzed the Ser9 phosphorylation of GSK3β as a measure of inhibition. We found that the stressed mice (CUMS) presented decreased Ser9 phosphorylation levels when compared with CNT mice, both in the prefrontal cortex (Figure 1B) and in the hippocampus (Figure 1C). These results confirm that chronic stress induces the hyperactivity of GSK3β. Then, we wanted to test if meridianins could inhibit GS3Kβ in vivo in the different brain regions related to depression such as the prefrontal cortex (PFC), the hippocampus (Hipp), the nucleus accumbens (NAcc), and the amygdala. To this end, we injected meridianins (500 nM) in the third ventricle of the brain of adult (3-month-old) mice. This dose was selected based on our previous works in which we observed it was the most efficient one [31,54]. We then sacrificed the mice at different time points to evaluate GSK3β inhibition (Figure 1D). We found GSK3β inhibition at 20 min and 1 h after injection in the prefrontal cortex (PFC) (Figure 1E) and in the hippocampus (Figure 1F). Conversely, we only found GSK3β inhibition at 20 min in the amygdala (Figure 1G), and in the NAcc, we found just a trend (Figure 1H). These results confirm that the intraventricular injection of meridianins (500 nM) inhibits GSK3β in different brain areas, with the strongest effects in the hippocampus and the PFC 20 min after injection.

### 2.2. Meridianins Modulate Molecular Pathways Involved with GSK3β Signaling

Once we observed that meridianins could modulate GSK3β activity in vivo, we then aimed to know if this inhibition was accompanied by changes in other proteins that are known to be involved with GSK3β signaling. To this end, we performed Western blot experiments in lysates from both the hippocampus and the PFC lysates of those mice that were treated with meridianins for 20 min and their respective controls. First, we evaluated the levels of kinases that are described as GSK3β modulators such as Akt, PKC, and PKA [10,11]. To check for Akt activity, we looked at phospho-Akt^Ser473^ levels and found that their levels were significantly increased in the PFC of those mice treated with meridianins for 20 min (Figure 2A) but not in the hippocampus when compared with controls (Figure 2B). Then, we looked for PKC activity by using an antibody that detects the phosphorylation of proteins at phospho-Ser PKC substrate motifs. We found no changes between the groups in the PFC (Figure 2C). Contrarily, we found increased activity of PKC in the hippocampus in the mice treated with meridianins with respect to the mice treated with vehicle (Figure 2D). Next, we analyzed the levels of PKA activation. To this end, we used an anti-phospho-PKA substrate antibody that detects the proteins containing phosphorylated Ser/Thr residue within the consensus sequence for PKA, giving us a readout of PKA activity. Interestingly, we found an increase in PKA activity in PFC (Figure 2E) but not in the hippocampus in the mice treated with meridianins, compared with the mice treated with vehicle (Figure 2F). These results suggest different mechanisms of action of meridianins depending on the brain region. One of the substrates under PKA signaling associated with depression and antidepressant responses is CREB [9,11]. We looked at the levels of phospho-CREB^Ser133^ in both brain regions and found no changes in the hippocampus or the PFC when comparing both groups. Lastly, we assessed the levels of a major phosphorylated substrate in terms of PKC activity, the so-called glutamate receptor subunit GluR1. We evaluated the GluR1 activity using an antibody that detects the phosphorylation of the receptor at Ser 831. We found increased phospho-GluR1^Ser831^ levels in the mice treated with meridianins, compared with the mice treated with vehicle, specifically in the hippocampus. These results suggest that meridianins can also modulate synaptic transmission, especially in this brain region.

### 2.3. Meridianins Increase Spontaneous Synaptic Activity in the Hippocampal CA1

We observed that meridianins exert GSK3β inhibition in brain regions such as the PFC and the hippocampus. Moreover, we also found that meridianins increased the activity of the excitatory glutamate receptor subunit GluR1 in the hippocampus. To further examine if this modulation was accompanied by electrophysiological changes, we assessed whether meridianins modulate hippocampal synaptic transmission in brain slices. We thus measured the neuronal spike rate and bursting in the pyramidal CA1 and CA3 and in the granular DG subfields using a multielectrode array (MEA) (Figure 3B). We found a meridianin-dependent increase in the spike rates of the pyramidal CA1 cells when compared with baseline (Figure 3C,D), but no meridianin-dependent effects were observed in the spike rates of pyramidal CA3 or granular DG layers (Figure 3C,D). We found no differences in the burst parameters (data not shown) in any of the three hippocampal areas analyzed. These results support the idea that meridianins have a moderate but specific effect on synaptic transmission or neuronal excitability in the hippocampal CA1.

### 2.4. Meridianins Improve Behavioral Deficits Induced by CUMS

To study the possible role of GSK3β inhibition in an MDD-like mouse model induced by chronic stress, we used the CUMS protocol. Several studies have already shown that CUMS induces important alterations in mice such as loss of body weight, increased anxiety, or cognitive decline [55,56]. To this end, we implanted adult (10-week-old) mice with osmotic minipumps to continuously deliver meridianins or saline (VEH) in the lateral ventricle for 28 days (Figure 4A). After surgery, the mice were randomly distributed to CNT or CUMS groups. The CUMS protocol lasted 28 days (Figure 4B). During the CUMS, we analyzed different behavioral tasks. First, in the middle of the protocol (day 13), we measured body weight and observed that only the stressed mice without treatment presented decreased body weight, compared with CNT VEH (Figure 4C). Then, we performed an open-field test to measure locomotor activity and anxiety. The results showed that all the groups presented similar levels of covered distance (Figure 4D). We then evaluated anxiety-like behavior by measuring the distance traveled in the center of the arena (Figure 4E). We found that CUMS VEH mice spent less time in the center of the arena in comparison with CNT VEH mice, and this was totally recovered in those stressed mice treated with meridianins (CUMS MER) (Figure 4E). Moreover, we evaluated the parallel index (1.0 means walking straight) as a measure of navigation strategies in the mice (Figure 4F). We found that CUMS VEH mice presented alterations in spatial navigation in comparison with control mice, and this was recovered in those stressed mice that received meridianins (CUMS MER) (Figure 4F). We then measured behavioral despair using the forced swim test. The results indicated that both groups of stressed mice spent more time floating when compared with CNT mice. Finally, we performed the novel object location task (NOLT) to evaluate spatial memory. The results indicated that CNT mice were able to distinguish between the old and the new object position. Conversely, CUMS VEH mice could not differentiate the new location of the object. However, these deficits were totally rescued in those mice treated with meridianins (CUMS MER) (Figure 4H). These results suggest that meridianins can improve some of the deficits induced by chronic stress such as body weight loss, anxiety, navigation, and spatial memory.

## 3. Discussion

Here, we showed that chronic treatment with meridianins in a mouse model of depression induced by chronic stress has beneficial effects. We first confirmed that chronic stress induces the hyperactivation of GSK3β. Then, we verified that meridianins can regulate in vivo GSK3β activity in different brain areas related to stress-related pathologies such as major depression disorder (MDD). We also showed that this modulation was accompanied by changes in the activity of proteins involved with GSK3β signaling or with changes in synaptic transmission. We then used a stress protocol to induce a depressive-like phenotype in mice and combined the stress protocol with meridianin administration. Interestingly, we found that some of the sequels induced by chronic stress were prevented with the meridianin treatment.

First, we confirmed that intraventricular meridianin administration can inhibit GSK3β in different brain areas. In this study, we focused on those brain regions that are known to play important roles in chronic stress and have been associated with GSK3β dysregulation. The hippocampus is one of the regions most affected by stress. Several studies reported alterations in the volume [57,58] and hippocampal-related cognitive tasks [59,60] in MDD. Moreover, changes in the activity of GSK3β in this region have been found in stress-related pathologies [16,61]. The prefrontal cortex (PFC) has emerged as another highly impaired brain region in MDD [62]. In addition, several groups have described alterations in the activity of GSK3β in this brain region [15,63]. These alterations have also been reported in the NAcc, where some groups have demonstrated that genetic GSK3β inhibition during a stress protocol is beneficial for mice [14]. Finally, recent data postulate the amygdala as a core structure in MDD [64,65], but little is known about GSK3β role in the amygdala in mood disorders. All these data suggest that GSK3β plays a crucial role in the pathophysiology of stress-related disorders including MDD. For this reason, its inhibition in different brain areas using meridianins could be a promising therapeutical approach.

An important question that is not completely understood is which GSK3β substrates could mediate its effect on mood regulation. One of the major regulators of GSK3β is Akt. Some studies reported decreased activity of Akt in depressed patients [15] and in rodent models of depression [66,67]. In our study, we observed that meridianin administration increased the Akt activity in the PFC, showing a similar effect as other antidepressants [68,69]. Moreover, we also found increased levels of phosphorylated PKA substrates in the PFC. Contrarily, this increase was not accompanied by changes in the levels of phosphorylated CREB. This could suggest that other PKA targets could be mediating GSK3β effects in the PFC. In contrast to the PFC, we saw increased levels of PKC phosphorylated substrates in the hippocampus that were accompanied by increased phosphorylation of GluR1 at Ser831. This is in line with previous research that points out PKC as one of the main kinases phosphorylating GluR1 specifically at this site [70,71]. This phosphorylation has been linked to enhanced synaptic plasticity and long-term potentiation (LTP) [72]. Moreover, studies using mice lacking this phosphorylation showed deficits in hippocampal tasks such as spatial learning [73]. This supports the idea that the GluR1 phosphorylation at specific Ser residues is crucial for memory formation and that meridianins can modulate the GluR1 activity. In this line, alterations in the levels and function of GluR1 have been already described in patients with mood disorders [74] and in mouse models of depression [75,76].

To confirm that meridianins can modulate synaptic activity, we performed electrophysiological recordings in acute hippocampal slices. Several studies reported synaptic activity alterations after CUMS [77] and in other mouse models of stress [78,79]. These alterations are often associated with cognitive deficits and with changes in the density of dendritic spines [77]. Here, we found an increase in synaptic performance specifically in the CA1 after meridianin administration that could be beneficial in a disease context such as MDD. Indeed, the GuR1 function is crucial for CA1 synaptic activity [72].

Then, we used the chronic unpredictable mild stress protocol (CUMS) to induce a depressive-like phenotype. We observed that meridianin administration was able to ameliorate some of the important sequelae induced by chronic stress. Those mice treated with meridianins presented a rescue in body weight loss and decreased anxiety levels, suggesting a role of meridianins in the emotional-like symptoms induced by chronic stress. We saw a clear effect of meridianins in cognition that could be associated with an enhancement of synaptic transmission. Although this could be due to an increase in the phosphorylation of the GluR1 receptor [72], we cannot rule out other possibilities [59,76].

Finally, this study presents some limitations. First, future work should clarify whether all the meridianins in the mixture, or only some of them, or perhaps some of them acting synergistically, are the directly responsible molecules for the observed effects. These could be achieved by using synthesized molecules, although not all the meridianins have been synthesized so far. Synthesis has been achieved for some meridianins and analogs through different pathways [80,81,82,83,84,85,86,87]. Moreover, although we demonstrated GSK3β inhibition by meridianins, we cannot exclude the possibility of alternative mechanisms of action that could be mediating the improvements observed in the stressed mice when treated with meridianins. Future studies should be focused also on elucidating these limitations. All in all, our results suggest that meridianins improve some of the deficits induced by chronic stress possibly via the inhibition of GSK3β.

## 4. Materials and Methods

### 4.1. Animals

For this study, we used male adult C57BL/6JOlaHsd mice (Envigo and Charles River). All mice were housed together until starting the chronic stress procedure in groups of four or five mice per cage. All the animals were housed with access to food and water ad libitum in a colony room kept at 19–22 °C and 40–60% humidity, under an inverted 12:12 h light/dark cycle (from 08:00 to 20:00). All animal procedures were approved by local committees (Universitat de Barcelona, CEEA (10141) and Generalitat de Catalunya (DAAM 315/18)), in accordance with the European Communities Council Directive (86/609/EU).

### 4.2. Marine Molecules

Marine compounds were obtained from the available sample collections at the University of Barcelona (BEECA Department, Faculty of Biology) from previous Antarctic projects (BLUEBIO, CHALLENGE). Briefly, the collected Antarctic marine organisms of the species *Aplidium falklandicum* were extracted with organic solvents, and the extracts were further purified through chromatographic methods (HPLC), as previously reported [36,37]. In the current work, we used the meridianins isolated from these previous reports, which were kept frozen at −20 °C until used. In our assays, we used the mixture of several meridianins (A–G) since the total sample amount was low, and it was not possible to identify which meridianins were present in the mixture, nor in which proportions, since this is a quite complex group of compounds [35].

### 4.3. Experimental Design

For the experiment shown in Figure 1D–H and Figure 2, we performed the following experimental design: Briefly, 25 male adult mice were stereotaxically injected (see Section 4.4) with meridianins and sacrificed after 20 (6 mice), 60 (7 mice), or 180 (6 mice) min. For the experiment shown in Figure 3, 9 adult mice were used for electrophysiological field recordings. For the experiments shown in Figure 1A,B and Figure 4, between 16 and 20 male mice were used per condition.

### 4.4. Immunoblot Analysis

Hippocampal samples were collected in cold lysis buffer containing 50 mM Tris base (pH 7.5), 10 mM EDTA, 1% Triton X-100, and supplemented with 1 mM sodium orthovanadate, 1 mM phenylmethylsulphonyl fluoride, 1 mg/mL leupeptin, and 1 mg/mL aprotinin. The samples were centrifuged at 16,000× *g* for 15 min, and the supernatants were collected. After incubation (1 h) in blocking buffer containing 2.5% BSA and non-fat powdered milk in Tris-buffered saline Tween (TBS-T) (50 mM Tris–HCl, 150 mM NaCl, pH 7.4, 0.05% Tween 20), the membranes were blotted overnight at 4 °C with primary antibodies. The antibodies used for immunoblot analysis were phospho-GSK3β at Ser9 (1:1000; Cell Signaling, #9336xz), GSK3β (1:1000; Cell Signaling, #9315), phospho-Akt Ser 473 (1:1000; Cell Signaling, #4060S), Akt (pan) (1:1000; Cell Signaling, #4691), phospho-Ser PKC substrates (1:1000; Cell Signaling, #2261), phospho-PKA substrates (RRXS*/T*) (1:1000; Cell Signaling, #9624), phospho-CREB (Ser 133) (1:1000; Millipore, #06-519), CREB (1:1000, Cell Signaling, #9197S), phospho-GluR1 at Ser 831 (1:1000; Millipore, #04-823), GluR1 (1:1000, Millipore, #ABN241), and GAPDH (1:1000; Millipore, #MAB374).

The membranes were then rinsed three times with TBS-T and incubated with horseradish peroxidase-conjugated secondary antibody for 1 h at room temperature. After washing with TBS-T, the membranes were developed using an enhanced chemiluminescence (ECL) kit (Santa Cruz Biotechnology, Dallas, TX, USA). The ImageLab densitometry program (ImageLab from ChemiDoc system from Bio-Rad) was used to quantify the different immunoreactive bands relative to the intensity of the α-GAPDH or, in the case of phospho-GSK3β, it was relativized with respect to total GSK3β.

### 4.5. Stereotaxic Surgery

For Figure 1 and Figure 2, C57BL/6JOlaHsd mice were deeply anesthetized with isoflurane (2% induction, 1.5% maintenance) and 2% oxygen and placed in a stereotaxic apparatus for injection into the third ventricle with 3 μL of 500 nM meridianins and were sacrificed 20 min, 1 h or 3 h after administration. For Figure 4, C57BL/6JOlaHsd mice were deeply anesthetized with isoflurane (2% induction, 1.5% maintenance) and 2% oxygen and placed in a stereotaxic apparatus for osmotic minipump (model 1002; Alzet, Palo Alto, CA, USA) implantation. A brain infusion kit (#0008663) was also used to deliver into the lateral (left) ventricle 0.11 μL per hour of the vehicle or 500 nM meridianins (0.1 mm posterior to bregma, ±0.8 mm lateral to the midline, and −2.5 mm ventral to the parenchyma surface). Cannulas were fixed on the skull with Loctite 454 (from Alzet). Minipumps, previously equilibrated overnight at 37 °C in PBS, were implanted subcutaneously in the back of the animal. After recovery, the mice were distributed between CNT and CUMS groups.

### 4.6. Electrophysiological Field Recordings

The brain coronal sections of 20-week-old male mice were obtained at 350 μm thickness on a vibratome (Microm HM 650 V, Thermo Scientific, Waltham, MA, USA) in an oxygenated (95% O_2_, 5% CO_2_) ice-cold artificial cerebrospinal fluid (aCSF). The sections were transferred to an oxygenated 32 °C recovery solution for 15 min, as previously described [88], and then to oxygenated aCSF, where they were incubated at room temperature for at least 1 h before electrophysiological field recording.

After recovery, the slices were transferred into 60MEA200/30iR-ITO MEA recording dishes and fully submerged in oxygenated aCSF at 37 °C. For the recording of spontaneous activity, the hippocampal formation slice surface was placed on MEA 60 planar electrodes arranged in an 8 × 8 array with the assistance of a digital camera. Raw traces were sampled at 5 kHz and recorded for 5 min from 58 electrodes simultaneously. To determine the effects of meridianins in the hippocampal spontaneous activity, after a 3 min baseline recording, the slices were incubated with 500 nM meridianins in aCSF [54] for 20 min. Then, spontaneous activity was recorded for an additional 3 min.

Spikes were identified and quantified as previously described [89]. In brief, we applied a high-pass filter with a 200 Hz Butterworth 2nd-order filter. We then assessed the noise level by using the signal standard deviation on each electrode and identified the spikes as currents with slope values between 0.2 and 1 and a negative amplitude larger than −20 mV. We applied the max interval method [90] to quantify the burst activity with the following parameter values: maximum ISI beginning and end, 200 ms and 200 ms, respectively; minimum burst duration 20 ms; minimum number of spikes in a burst, 5; and minimum interburst interval, 20 ms. We used MC Rack from Multi Channel Systems software for recording and signal processing. We selected the electrodes specifically positioned on different fields of the hippocampal formation by the image taken with a digital camera (Figure 3B).

### 4.7. Chronic Stress

One day before the beginning of the CUMS procedure described previously [91], the mice were individually housed and maintained isolated for the entire experiment. The CUMS procedure followed a random weekly schedule of commonly used mild stressors (one per day): restrain (1 h), food or water deprivation (24 h), home cage inclination (1 h), forced swimming (5 min), exposition to rat sawdust (4 h), and alterations in the light–dark cycle (24 h). The detailed stressor used each day is described in Appendix A. The CUMS protocol lasted 28 days and

### 4.8. Behavioral Test

Open-field (OF) and Novel object location test (NOL): For the novel object location test (NOL), an open-top arena (40 × 40 × 40 cm) with visual cues placed in the inner part of the walls was used. The mice were first habituated to the arena (1 day, 15 min). We considered this first exposition to the open arena as an open-field paradigm. We monitored the total traveled distance and time spent in the center of the arena as measures of locomotor activity and anxiogenic behavior, respectively. On day 2, two identical objects (A1 and a2) were placed in the arena. The mice were allowed to explore the objects for 10 min. Exploration was considered when the mouse sniffed the object. Then, 24 h later (D3), one object was moved from its original location to the diagonally opposite corner, and the mice were allowed to explore the arena and the object for 5 min. At the end of each trial, defecations were removed, and the apparatus was cleaned with 30% ethanol. Animal tracking and recording were performed using the automated SMART junior software (Panlab, Barcelona, Spain).

*Forced swim test:* A forced swim test was used to evaluate behavioral despair. The animals were subjected to a 6 min trial during which they are forced to swim in an acrylic glass container (35 cm height × 20 cm diameter) filled with water and from which they could not escape. The time that the test animal spent in the cylinder without making any movements except those required to keep its head above water was measured.

All the tests were conducted during the light cycle, and all the mice were randomized throughout the day. Only one test was conducted per day.

### 4.9. Statistics

Sample sizes were determined by using the power analysis method: 0.05 alpha value, 1 estimated sigma value, and 75% of power detection. All data are expressed as mean  ±  SEM. Normal distribution was tested with the d’Agostino and Pearson omnibus normality test. If the test was passed, statistical analysis was performed using parametric statistical analysis. In the experiments with normal distribution, statistical analyses were performed using an unpaired two-sided Student’s *t*-test (95% confidence), one-way ANOVA, and two-way ANOVA followed by Dunnett’s or Tukey’s post hoc tests, as appropriate and indicated in the figure legends. The values of *p*  <  0.05 were considered statistically significant. Grubbs and ROUT tests were performed to determine the significant outlier values. All statistical analyses were carried out using GraphPad Prism software version 8.0.2 for Windows (GraphPad Software, San Diego, CA, USA, www.graphpad.com, accessed on 6 July 2022).

## 5. Conclusions

In conclusion, in the present work, we showed that meridianins inhibit GSK3β in vivo in several brain regions involved with the pathogenesis of MDD. Furthermore, we deepened the understanding of the molecular pathways underlying GSK3β inhibition by meridianins such as PKA, PKC, and GluR1 and also the synaptic consequences of GSK3β inhibition by meridianins in the hippocampus. We finally showed that intraventricularly delivered meridianins ameliorate several sequelae induced by chronic stress.

## Figures and Tables

**Figure 1 marinedrugs-20-00648-f001:**
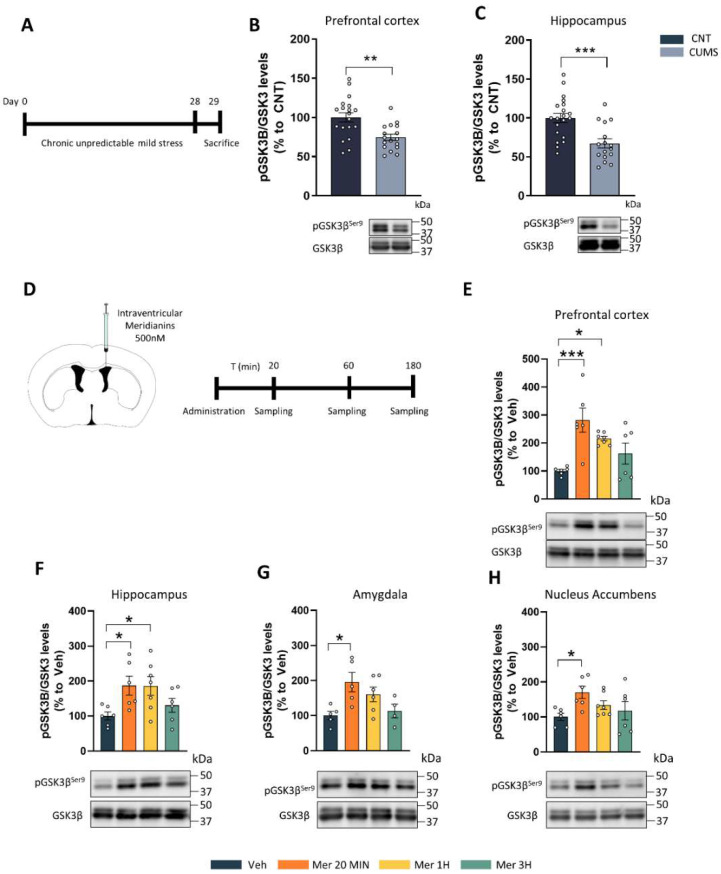
GSK3β phosphorylation levels upon chronic stress and its inhibition in different brain regions: (**A**) schematic representation of the experimental design. Ten-week-old C57BL6/J mice were subjected to the chronic unpredictable mild stress (CUMS) protocol for 28 days. One day after the last stressor, mice were sacrificed; (**B**) densitometry quantification and representative immunoblots of phospho-GSK3βSer9 levels in the prefrontal cortex. Unpaired *t*-test: t = 3.473, df = 34; (**C**) densitometry quantification and representative immunoblots of phospho-GSK3βSer9 levels in the hippocampus. Unpaired *t*-test: t = 3.835, df = 43. ** *p* < 0.005, *** *p* < 0.0005 compared with CNT; (**D**) three-month-old C57BL6/J male mice were stereotaxically injected into the lateral (left) ventricle with 3 μL of 500 nM meridianins and were sacrificed 20 min, 1 h or 3 h after administration; (**E**) densitometry quantification and representative immunoblots of phospho-GSK3βSer9 levels in the prefrontal cortex. One-way ANOVA: F(3,21) = 7.481, *p* = 0.0014; (**F**) densitometry quantification and representative immunoblots of phospho-GSK3βSer9 levels in the hippocampus. One-way ANOVA: F(3,21) = 3.568, *p* = 0.0314; (**G**) densitometry quantification and representative immunoblots of phospho-GSK3βSer9 levels in the amygdala. One-way ANOVA: F(3,16) = 4,191, *p* = 0.0228; (**H**) densitometry quantification and representative immunoblots of phospho-GSK3βSer9 levels in the NAcc. One-way ANOVA: F(3,21) = 2.843, *p* = 0.0623. Data are mean ± SEM. Dunnett’s test as a post hoc analysis was used. * *p* < 0.05; *** *p* < 0.005; compared with Veh mice. Molecular weight markers’ positions are indicated in kDa (n = 5–7 mice/group). Veh = vehicle and Mer = meridianins.

**Figure 2 marinedrugs-20-00648-f002:**
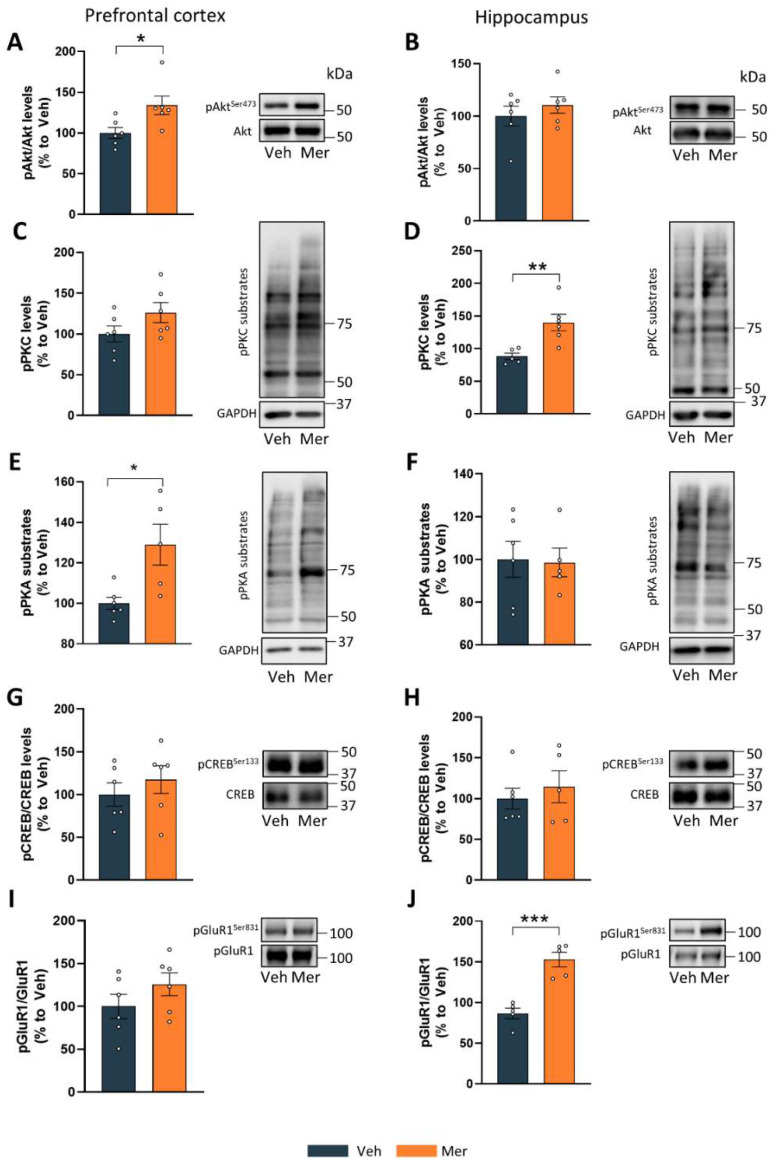
Biochemical assessment of mice treated with meridianins. Densitometry quantification and representative immunoblots of phosphorylated Akt at Ser 473 in the prefrontal cortex (**A**) (t = 2.569, df = 10, *p* < 0.05) and in the hippocampus (**B**) (t = 0.8531, df = 10, *p* = 0.04). Densitometry quantification and representative immunoblots of phosphorylated PKC substrates in the prefrontal cortex (**C**) (t = 1.661, df = 10, *p* = 0.13) and in the hippocampus (**D**) (t = 3.506, df = 9, *p* < 0.05). Densitometry quantification and representative immunoblots of phosphorylated PKA substrates in the prefrontal cortex (**E**) (t = 2.984, df = 9, *p* < 0.05) and in the hippocampus (**F**) (t = 0.1293, df = 9, *p* = 0.9). Densitometry quantification and representative immunoblots of phosphorylated CREB at Ser 133 in the prefrontal cortex (**G**) (t = 0.8204, df = 10, *p* = 0.43) and in the hippocampus (**H**) (t = 0.6394, df = 9, *p* = 0.5385). Densitometry quantification and representative immunoblots of phosphorylated GluR1 at Ser 831 in the prefrontal cortex (**I**) (t = 1.326, df = 10, *p* = 0.21) and in the hippocampus (**J**) (t = 6.028, df = 8, *p* < 0.005). Data are mean ± SEM. Two-tailed Student’s *t*-test was used. * *p* < 0.05; ** *p* < 0.05; *** *p* < 0.005 compared with Veh mice. Molecular weight markers’ positions are indicated in kDa in (**B**) (n = 5–6 mice/group). Veh = vehicle and Mer = meridianins.

**Figure 3 marinedrugs-20-00648-f003:**
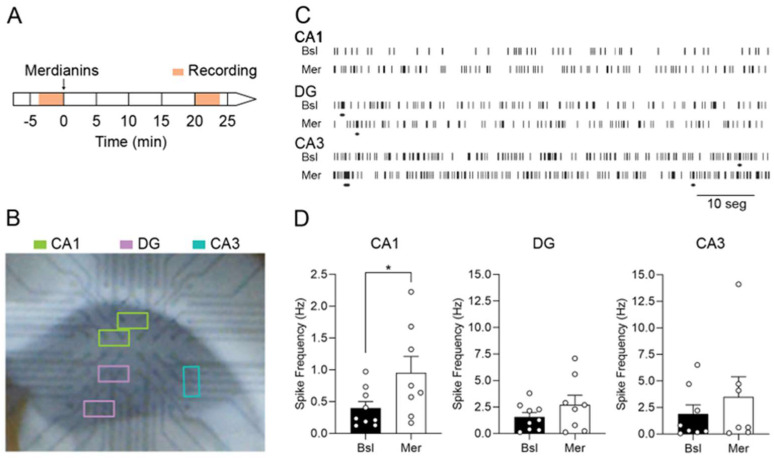
Meridianins specifically increase synaptic activity in pyramidal CA1 neurons: (**A**) schematic diagram of the experimental multielectrode array (MEA) recording timeline; (**B**) illustrative image of a brain coronal slice on the MEA (magnification). Recordings from the selected electrodes located on CA1, DG, and CA3 (in green, purple, and blue, respectively) were analyzed; (**C**) illustrative timescale spike raster of CA1, DG, and CA3 spontaneous activity in basal (Bsl) conditions and after incubation with 500 nM meridianins (Mer). The horizontal lines under each raster define bursts; (**D**) graphs show quantification of spike frequency in basal conditions and after treatment with meridianins in CA1 (t = 2.568, *p* = 0.0371), DG (t = 1.944, *p* = 0.0930), and CA3 (t = 1.449, *p* = 0.1975). Data are presented as mean ± SEM. * *p* < 0.05, two-tailed Student’s *t*-test. (n = 7–8 mice/group).

**Figure 4 marinedrugs-20-00648-f004:**
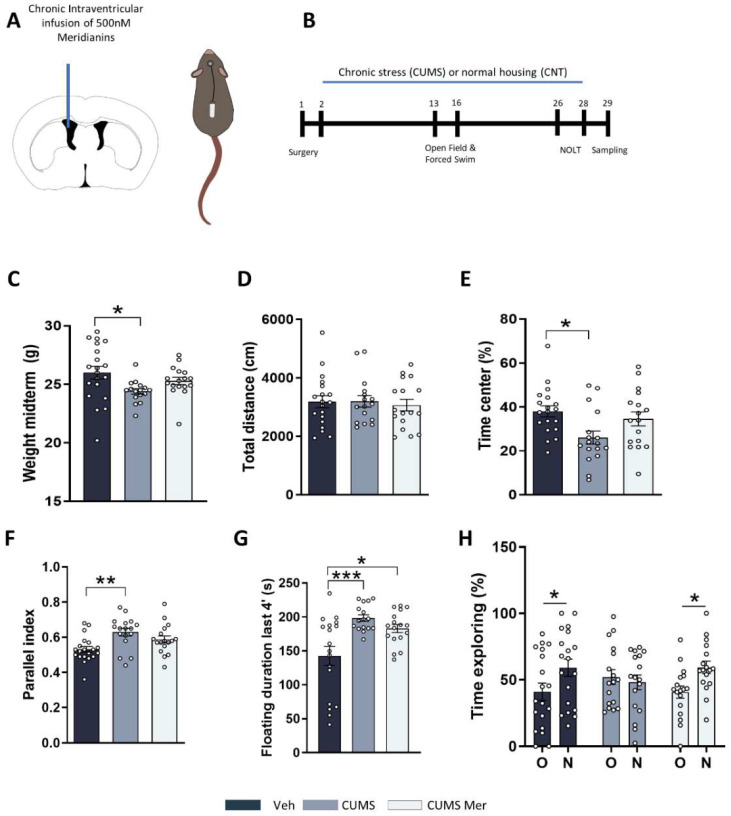
Effects of intraventricular meridianins’ chronic delivery in stressed mice: (**A**) schematic representation of the experimental design. Ten-week-old male WT mice were chronically treated for 28 days with 500 nM meridianins or vehicle delivered into the lateral (left) ventricle; (**B**) during the stress protocol, several behavioral tasks were performed; (**C**) body weight changes induced by chronic stress. One-way ANOVA: F(2,50) = 3.716, *p* = 0.013. In the open field, locomotor activity (**D**), time spent in the center (**E**), and parallel index (**F**) were monitored for 15 min. Locomotor activity, one-way ANOVA, F(2,49) = 0.1206, *p* = 0.8866. Time in center, one-way ANOVA, F(2,50) = 4.519, *p* = 0.0157. Parallel index, F(2,51) = 6.441, *p* = 0.0032; (**G**) in the forced swim test, floating duration was analyzed for the last 4 min of the test. One-way ANOVA, F(2,48) = 9.128, *p* = 0.0004; (**H**) in the novel object location task, spatial memory was evaluated 24 h after a training trial as the percentage of total time spent exploring either the object placed at a new location (N) or the object placed at the old location (O). Two-way ANOVA, new location effect, F(1,100) = 5.460, *p* = 0.0214. * *p* < 0.05 CNT; ** *p* < 0.005; *** *p* < 0.0005. N = 17–20 mice per group. Veh = vehicle and Mer = meridianins.

## Data Availability

The datasets generated during and/or analyzed during the current study are available from the corresponding author upon reasonable request.

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
