# Peer review of "Meridianins Inhibit GSK3β In Vivo and Improve Behavioral Alterations Induced by Chronic Stress"

_marinedrugs, 2022, doi:10.3390/md20100648_

Round 1

Reviewer 1 Report

Title: Meridianins Inhibit GSK3β In Vivo and Improve Cognitive and Emotional Alterations Induced by Chronic Stress

By Sancho-Balsells et al

Submitted to Marine drugs

This work is a research study on the potential protective effect of meridianins against MDD caused by chronic stress and the involvement of major molecular pathways in brain areas. This research has certain value, however, there still some issues that should be addressed. 

Major comments:

1.     A detailed experimental design must be included before “4.3. Immunoblot analysis” section, line 314 to outline the whole experiments performed in this work. In the first invivo experiment, what is the total number of mice used for this experiment? The figure 1A is confusing regarding the timeline, this should be modified to indicate the treatment protocol for each group.

2.     It is very important to investigate the effect of meridianins on GSK3β activity and signaling and synaptic activity in mice exposed to CUMS.

3.     Why were the open field and forced swim tests not performed by the end of experiment as NOLT? Is the 2 weeks period from CUMS protocol enough for mice to develop anxiety and depression symptoms? 

Minor comments:

1.     English language needs improvement.

2.     The results should be presented in the abstract in a better way. 

3.     In line 121, “These results confirm that intraventricular injection of and inhibits GSK3β”, please check the accuracy of this sentence. 

4.     In line 311, indicate the reference number of the Antarctic projects.

5.     In line 334 and 335, please indicate all phosphorylated proteins that were interpreted with respect to their total proteins.  

6.     In line 380, please replace the word elsewhere by previously.

7.     In line 387-398, please indicate the days used with a unified method according to the exact day number on the timeline of the experiment. 

8.     In the statistics section, please indicate the software used to conduct statistical analysis. 

Author Response

This work is a research study on the potential protective effect of meridianins against MDD caused by chronic stress and the involvement of major molecular pathways in brain areas. This research has certain value, however, there still some issues that should be addressed.

Major comments:

  1. A detailed experimental design must be included before “4.3. Immunoblot analysis” section, line 314 to outline the whole experiments performed in this work. In the first in vivo experiment, what is the total number of mice used for this experiment? The figure 1A is confusing regarding the timeline, this should be modified to indicate the treatment protocol for each group.

We agree with the reviewer that a section explaining the experimental design in detail was missing. We have now added this section (Current section 4.3.) before the immunoblot analysis section (Current section 4.4.). Additionally, we have modified the timeline in Figure 1A to make it easier to understand.

  1. It is very important to investigate the effect of meridianins on GSK3β activity and signaling and synaptic activity in mice exposed to CUMS.

We agree with the reviewer that this information was missing. To accomplish with the reviewer request we initially analyzed the effect of stress and meridianins on GSK3β. To do so, we performed Western Blot analysis. We first found that those stressed mice (CUMS VEH) presented an hyperactivation of GSK3β (See Figure 1B and 1C). This result goes in line with the literature and support our hypothesis. We then studied the effect of meridianins in those stressed mice. Contrary to what was expected, we could not find any difference between those CUMS mice treated with VEH or treated with MER (See attached figure below). We then analyzed signaling pathways downstream GSK3β and we could not find differences between any group (See attached rebuttal figure 1 below). We concluded that extensive manipulation (Invasive surgeries + long-term behavioral experimentation + the presence of an intra-cerebral mini-pump for four weeks) have altered, probably, the phosphorylation levels of GSK3B and its downstream signaling in a uncontrollable fashion. Indeed, in our team we have widely observed that mice with extensive manipulation have severe alterations in many molecular parameters that in basal states are changed. One example from our own lab in this sense has been recently published

 Fernández-García S, Conde-Berriozabal S, García-García E, Gort-Paniello C, Bernal-Casas D, García-Díaz Barriga G, López-Gil J, Muñoz-Moreno E, Soria G, Campa L, Artigas F, Rodríguez MJ, Alberch J, Masana M. M2 cortex-dorsolateral striatum stimulation reverses motor symptoms and synaptic deficits in Huntington's disease. Elife. 2020 Oct 5;9:e57017. doi: 10.7554/eLife.57017.

In this previous publication [1] we have observed that in a muse model of Huntington’s disease, upon massive manipulation, it losses its main biochemical hallmarks such as altered levels of DARPP32, BDNF, NMDAR, etc.

Rebuttal figure 1 (See attached document). These results are only depicted in the rebuttal letter for reviewer commodity. As stated below, we do not think that they help to improve the manuscript and that is why we put them only here

Also, we performed the sampling 24h after the last day of treatment (minipumps last for 28 days and we performed brain sampling at day 29 post mini-pump implantation). Looking at the acute effects depicted in figure 1 regarding to GSK3β inhibition (from 20 to 60 minutes), it is conceivable that this timing could also exert some effect on this apparent lack of GSK3β inhibition and the subsequent changes on its downstream signaling.

Thus, we have not included this negative result because could be confusing and, from our point of view, it is not providing useful information to the manuscript and, furthermore, we already showed modulation of GSK3β in vitro [2] and in vivo (Figure 1 of the present manuscript).

Finally, another possibility is that meridianins could also act through other mechanisms. In this sense, in the current version of the manuscript we have included a “weaknesses paragraph” at the end of the discussion section and we have stated that “although we demonstrated a GSK3β inhibition by meridianins, we cannot exclude the possibility of alternative mechanisms of action that could be mediating the improvements observed in stressed mice when treated with meridianins. Future studies should be focused on elucidating the targets altered by stress that are modulated by meridianins” (page 9, lines 307-312).

Finally, we sincerely thank to the reviewer for this important and useful observation.

  1. Why were the open field and forced swim tests not performed by the end of experiment as NOLT? Is the 2 weeks period from CUMS protocol enough for mice to develop anxiety and depression symptoms? 

This is a very important observation from the reviewer,

It has been previously reported that 2 weeks of stress has already an effect in mice behavior. There is a paper showing that with 2 weeks of CUMS mice already show alterations in the body weight [3]. Another study also demonstrated that 10 days of stress was sufficient to induce huge changes in mice such as decreased social interaction, increased anxiety and behavioural despair among others [4].

Also, in another project from the lab, we also observed that 13 days of stress is enough to observe some deficits in the forced swimming test as depicted below (Rebuttal figure 2). These results are only shown as supporting material for the reviewer since they belong to another project:

Rebuttal figure 2 (See attached document). These results are only depicted in the rebuttal letter for reviewer commodity. As stated above, they belong to another project and they only used to support the idea that up to two weeks of CUMS is enough to provoke behavioral sequels.

Finally, we performed some of the behavioural tasks during the second week to evaluate the effect of meridianins during the stress protocol. The main rationale is related with the dynamics of GSK3β inhibition. As illustrated in the previous point and as observed in figure 1, the inhibition of GSK3β lasts for 60 minutes at maximum and only in some brain regions. Therefore, we were afraid of losing the effect of this inhibition if we were performing the behavioral characterization just after CUMS (which would be behind the minipump treatment (28 days)) and during up to 1-2 weeks.

Minor comments:

  1. English language needs improvement.

We agree with the reviewer that the text needed language improvements. We have made some language changes to improve the manuscript throughout the text.

  1. The results should be presented in the abstract in a better way. 

We agree with the reviewer. We have modified the abstract to summarize better the results obtained (see changes in red).

  1. In line 121, “These results confirm that intraventricular injection of and inhibits GSK3β”, please check the accuracy of this sentence. 

We apologize for this grammar mistake. We have corrected it in the text. We thank to the reviewer for this observation.

  1. In line 311, indicate the reference number of the Antarctic projects.

The references for the projects are indicated in the Funding section. Nevertheless, we have included the names of the projects in the text though, as suggested (Lines 339 and 340).

  1. In line 334 and 335, please indicate all phosphorylated proteins that were interpreted with respect to their total proteins.  

This information is shown in the Y-axis of the graphs represented in Figure 2. Briefly, phospho-Akt, phospho-CREB and phospho-GluR1 were interpreted with respect to their total proteins.

  1. In line 380, please replace the word elsewhere by previously.

We have made the corresponding change in the text.

  1. In line 387-398, please indicate the days used with a unified method according to the exact day number on the timeline of the experiment. 

We agree with the reviewer that this part need further clarification. For this reason, we have added a Supplementary Table in which we describe the stressors used each day.

  1. In the statistics section, please indicate the software used to conduct statistical analysis. 

We agree with the reviewer that this information was missing. We have added this information at the end of the statistics section.

REFERENCES

  1. Fernández-García, S.; Conde-Berriozabal, S.; García-García, E.; Gort-Paniello, C.; Bernal-Casas, D.; Barriga, G.G.D.; López-Gil, J.; Muñoz-Moreno, E.; Soria, G.; Campa, L.; et al. M2 Cortex-Dorsolateral Striatum Stimulation Reverses Motor Symptoms and Synaptic Deficits in Huntington’s Disease. Elife 2020, 9, 1–24, doi:10.7554/ELIFE.57017.
  2. Llorach-Pares, L.; Rodriguez-Urgelles, E.; Nonell-Canals, A.; Alberch, J.; Avila, C.; Sanchez-Martinez, M.; Giralt, A. Meridianins and Lignarenone B as Potential GSK3β Inhibitors and Inductors of Structural Neuronal Plasticity. Biomolecules 2020, 10, doi:10.3390/BIOM10040639.
  3. Kuti, D.; Winkler, Z.; Horváth, K.; Juhász, B.; Szilvásy-Szabó, A.; Fekete, C.; Ferenczi, S.; Kovács, K.J. The Metabolic Stress Response: Adaptation to Acute-, Repeated- and Chronic Challenges in Mice. iScience 2022, 25, doi:10.1016/j.isci.2022.104693.
  4. Dang, R.; Wang, M.; Li, X.; Wang, H.; Liu, L.; Wu, Q.; Zhao, J.; Ji, P.; Zhong, L.; Licinio, J.; et al. Edaravone Ameliorates Depressive and Anxiety-like Behaviors via Sirt1/Nrf2/HO-1/Gpx4 Pathway. J. Neuroinflammation 2022, 19, 1–29, doi:10.1186/s12974-022-02400-6.
  5. Núñez-Pons, L.; Nieto, R.M.; Avila, C.; Jiménez, C.; Rodríguez, J. Mass Spectrometry Detection of Minor New Meridianins from the Antarctic Colonial Ascidians Aplidium Falklandicum and Aplidium Meridianum. J. Mass Spectrom. 2015, 50, 103–111, doi:10.1002/JMS.3502.
  6. Rodríguez-Urgellés, E.; Sancho-Balsells, A.; Chen, W.; López-Molina, L.; Ballasch, I.; Castillo, I. del; Avila, C.; Alberch, J.; Giralt, A. Meridianins Rescue Cognitive Deficits, Spine Density and Neuroinflammation in the 5xFAD Model of Alzheimer’s Disease. Front. Pharmacol. 2022, 13, doi:10.3389/FPHAR.2022.791666.
  7. Jope, R.; Roh, M.-S. Glycogen Synthase Kinase-3 (GSK3) in Psychiatric Diseases and Therapeutic Interventions. Curr. Drug Targets 2012, 7, 1421–1434, doi:10.2174/1389450110607011421.
  8. Del Ser, T.; Steinwachs, K.C.; Gertz, H.J.; Andrés, M. V.; Gómez-Carrillo, B.; Medina, M.; Vericat, J.A.; Redondo, P.; Fleet, D.; León, T. Treatment of Alzheimer’s Disease with the GSK-3 Inhibitor Tideglusib: A Pilot Study. J. Alzheimer’s Dis. 2013, 33, 205–215, doi:10.3233/JAD-2012-120805.
  9. Bhat, R. V.; Andersson, U.; Andersson, S.; Knerr, L.; Bauer, U.; Sundgren-Andersson, A.K. The Conundrum of GSK3 Inhibitors: Is It the Dawn of a New Beginning? J. Alzheimer’s Dis. 2018, 64, S547–S554, doi:10.3233/JAD-179934.

Reviewer 2 Report

In the present manuscript, the authors demonstrated that meridianins inhibit GSK3beta and improve the changes induced by chronic stress.

There were some concerns to proceed the manuscript for publication.

In Introduction, the text in the section for marine natural products and meridianins was written by citation from almost authors' own papers. Were not there any references written by other researchers?

In the present study, the authors injected meridianins in the third ventricle and the phosphorylation of proteins in PFC and Hipp was changed. How about the other injection sites? Did meridianins interpenetrate PFC and Hipp?

In Figure 4, figure legends were mismatched  to each A-H figure. (H) was lost in figure legends.

Moreover, in figure 4, why did not the authors perform CNT Mer group? And, how about the phosphorylation of GSK3beta alterations in this experiments? They should be studied if the authors demonstrated that meridianins improved cognitive and emotional alterations induced by chronic stress though inhibition of GSK3beta.

Meridianins are a group of marine-derived indole alkaloids which are reported to possess kinase inhibitory activities. There were 7 derivatives in meridianins. How many and how levels of meridianin derivatives were contained in the extracts used in the present study?

Author Response

Dear reviewer

We would like to re-submit our manuscript entitled "Meridianins Inhibit GSK3β In Vivo and Improve Behavioural Alterations Induced by Chronic Stress" by Sancho-Balsells et al. Our work is original research, it has not been previously published and it has not been submitted for publication elsewhere while under consideration. All authors declare there is no conflict of interest.

We thank the reviewers and the editors for their thoughtful comments. We have addressed all the questions that required new experimentation and changes in text sections. We honestly think that such changes strengthened the paper, and we indicate its exact new location in our point-by-point reply.

Please note that the modified text in the main text appears into the manuscript marked up in red.

REVIEWER 2

In the present manuscript, the authors demonstrated that meridianins inhibit GSK3beta and improve the changes induced by chronic stress.

There were some concerns to proceed the manuscript for publication.

In Introduction, the text in the section for marine natural products and meridianins was written by citation from almost authors' own papers. Were not there any references written by other researchers?

We agree with the reviewer that more citations were needed. We have added some more references now to include a wider number of authors from different laboratories (See references 25, 26, 27, 32, 33, 34, 38, 39, 41, 43, 44, 45, 47, 48, 49, 50, 51, 52, 53, 55, 56).

In the present study, the authors injected meridianins in the third ventricle and the phosphorylation of proteins in PFC and Hipp was changed. How about the other injection sites? Did meridianins interpenetrate PFC and Hipp?

We apologize for the unclear information. Here, we injected meridianins in the third ventricle (only one injection site). With this design meridianins reached all the brain regions taking advantage of the cerebrospinal fluid circulation. We first checked GSK3β inhibition by studying the phosphorylation level of GSK3β in different brain areas related with stress-induced sequelae namely, prefrontal cortex, hippocampus, amygdala and nucleus accumbens. As we saw the strongest effect of inhibition in the prefrontal cortex and hippocampus, we then focused with these two brain regions to study downstream pathways and regulators of GSK3β.

In Figure 4, figure legends were mismatched to each A-H figure. (H) was lost in figure legends.

We thank the reviewer for the correction. We have modified the figure legend accordingly.

Moreover, in figure 4, why did not the authors perform CNT Mer group? And, how about the phosphorylation of GSK3beta alterations in this experiments? They should be studied if the authors demonstrated that meridianins improved cognitive and emotional alterations induced by chronic stress though inhibition of GSK3beta.

We agree with the reviewer that some experiments were missing. Regarding the CNT Mer group we believe it was not necessary because:

  • We wanted to test the effect of meridianins in mice that have an hyperactivation of GSK3β. CNT mice do not present GSK3β hyperactivity, so we did not think that using an inhibitory treatment against GSK3β was interesting enough in the CNT group.
  • Related with the previous point: With this experimental design we think we are in line with the three Rs (Replacement, Reduction and Refinement) principle widely required in the scientific community in the context of animal use for experimentation and according to European Communities Council Directive (86/609/EU).
  • Finally, we did not have an unlimited amount of meridianins to perform the experiment. As mentioned in the methods section, meridianins are extracted from Aplidium ascidians located in Antarctica. The amount of meridianins needed for a 28-day treatment was very high to include another experimental group. Here we prioritized to have a huge sample size per group rather than including another experimental condition.

Regarding the phosphorylation of GSK3β in this experiment, we have already addressed this concern in the response/point 2 from Reviewer 1. We put it here in any case for the reviewer’s commodity:

“We agree with the reviewer that this information was missing. To accomplish with the reviewer request we initially analyzed the effect of stress and meridianins on GSK3β. To do so, we performed Western Blot analysis. We first found that those stressed mice (CUMS VEH) presented an hyperactivation of GSK3β (See Figure 1B and 1C). This result goes in line with the literature and support our hypothesis. We then studied the effect of meridianins in those stressed mice. Contrary to what was expected, we could not find any difference between those CUMS mice treated with VEH or treated with MER (See attached figure). We then analyzed signaling pathways downstream GSK3β and we could not find differences between any group (See attached figure). We concluded that extensive manipulation (Invasive surgeries + long-term behavioral experimentation + the presence of an intra-cerebral mini-pump for four weeks) have altered, probably, the phosphorylation levels of GSK3B and its downstream signaling in a uncontrollable fashion. Indeed, in our team we have widely observed that mice with extensive manipulation have severe alterations in many molecular parameters that in basal states are changed. One example from our own lab in this sense has been recently published

 Fernández-García S, Conde-Berriozabal S, García-García E, Gort-Paniello C, Bernal-Casas D, García-Díaz Barriga G, López-Gil J, Muñoz-Moreno E, Soria G, Campa L, Artigas F, Rodríguez MJ, Alberch J, Masana M. M2 cortex-dorsolateral striatum stimulation reverses motor symptoms and synaptic deficits in Huntington's disease. Elife. 2020 Oct 5;9:e57017. doi: 10.7554/eLife.57017.

In this previous publication [1] we have observed that in a muse model of Huntington’s disease, upon massive manipulation, it losses its main biochemical hallmarks such as altered levels of DARPP32, BDNF, NMDAR, etc.

Rebuttal figure 1 (See attachment). These results are only depicted in the rebuttal letter for reviewer commodity. As stated below, we do not think that they help to improve the manuscript and that is why we put them only here.

Also, we performed the sampling 24h after the last day of treatment (minipumps last for 28 days and we performed brain sampling at day 29 post mini-pump implantation). Looking at the acute effects depicted in figure 1 regarding to GSK3β inhibition (from 20 to 60 minutes), it is conceivable that this timing could also exert some effect on this apparent lack of GSK3β inhibition and the subsequent changes on its downstream signaling.

Thus, we have not included this negative result because could be confusing and it is not providing useful information to the manuscript and, furthermore, we already showed modulation of GSK3β in vitro [2] and in vivo (Figure 1 of the present manuscript).

Finally, another possibility is that meridianins could also act through other mechanisms. In this sense, in the current version of the manuscript we have included a “weaknesses paragraph” at the end of the discussion section and we have stated that “although we demonstrated a GSK3β inhibition by meridianins, we cannot exclude the possibility of alternative mechanisms of action that could be mediating the improvements observed in stressed mice when treated with meridianins. Future studies should be focused on elucidating the targets altered by stress that are modulated by meridianins” (page 9, lines 308-3012).

We sincerely thank to the reviewer for this important and useful observation.”

Meridianins are a group of marine-derived indole alkaloids which are reported to possess kinase inhibitory activities. There were 7 derivatives in meridianins. How many and how levels of meridianin derivatives were contained in the extracts used in the present study?

We agree with the reviewer that this information was missing in the manuscript. The extract used in our assays was a mixture because there was not enough amount of material to isolate all the different compounds in the mixture. Although there are 8 main meridianins (A to H; https://marinlit-rsc-org.sire.ub.edu/compounds?s=meridianin checked on August 16th, 2022), there are many other related meridianins and dimers in the extracts of these animals in minor amounts [5] and It is a very challenging task to purify them from the mixture. That’s the reason why we have used mixtures also in previous studies [6]. In order to expose these weaknesses, we have added a “limitation paragraph” in the discussion stating (page 10, lines 314-324) that:

“Finally, this study presents some limitations. First, future work should clarify whether all the meridianins in the mixture or only some of them, or perhaps some of them acting synergistically, are the direct responsible molecules for the observed effects. These could be done by using synthesized molecules, although not all the meridianins have been synthesized so far. Synthesis has been achieved for some meridianins and analogs through different pathways [80–87]. Moreover, although we demonstrated a GSK3β inhibition by meridianins, we cannot exclude the possibility of alternative mechanisms of action that could be mediating the improvements observed in stressed mice when treated with meridianins. Future studies should be focused on elucidating the targets altered by stress that are modulated by meridianins.”

Reviewer 3 Report

The authors investigated effects of meridianins (marine molecules from the Antarctic marine organisms) on the activity of GSK3β and related signalling pathways as GSK3β inhibitors are considered a promising therapeutic option in major depression disorder (MDD). They presented that meridianins act as a GSK3β inhibitors and modulators of PKA, PKC and Akt signalling in a brain region-specific manner. They also found that meridianins increased synaptic activity, specifically in the CA1 area of the hippocampus, probably due to the increased activity at the GluR1 subunit of the glutamate receptors. Finally, they showed that meridianins improve some chronic unpredictable mild stress (CUMS)-induced parameters such as weight loss, increased anxiety and cognitive impairment. The study is relevant and interesting, and the manuscript is easy to follow.

However, in my opinion, some additional experiments are needed before considering acceptance.

·      Meridianins are a very wide term including many potential active compounds. It is highly recommended to identify more precisely the active compound(s) from the extract.  Some meridianins (C,D,F,G) have been successfully synthesized (https://www.mdpi.com/1420-3049/27/7/2233/htm;).

·      It is already known that meridianins are potent kinase inhibitors, including the GSK-3 kinase (https://pubmed.ncbi.nlm.nih.gov/22512550/; https://www.mdpi.com/1660-3397/15/12/366/htm: https://pubmed.ncbi.nlm.nih.gov/32326204/;), and the authors have already shown that they are capable to improve cognitive deficits (https://www.ncbi.nlm.nih.gov/pmc/articles/PMC8908099/). In that way, many results presented here do not bring progress in the research field. The novelty here is related to meridianins and the MDD. Although the authors mentioned that „Future studies should be focused on elucidating the targets altered by stress that are modulated by meridianins ”, it is obligatory to study the effects of meridianins on GSK3β, PKA, PKC, Akt, CREB and GluR1 in the brain regions (prefrontal cortex, hippocampus, amygdala, nucleus accumbens) of the CUMS animals (and compare them with the effects observed in the CNT VEH mice) to confirm that the beneficial effects of meridianins in animal model od MDD are indeed GSK3β-related.

·      The study is performed with a single concentration of meridianins (500 nM). Usually, more than one dose is applied to monitor the dose-dependency of the activity.

·      Figure 1 (in vivo inhibition of GSK3β) presents results in healthy animals exposed to meridianins for a short period of time (20 min – 3 h). In general, it is visible that activity of GSK3β returns to control values after 3h in all brain areas. How is this acute effect relevant to the long-term therapeutic interventions that are assumed for MDD treatment? Why chronic treatment was not performed to see (if any) long-term effects of meridianins?  

Other comments

·      Chronic stress is used to model MDD. Throughout the manuscript, the authors are focused on MDD and meridianins as the potential therapeutics. However, in the title chronic stress is emphasized which is confusing. The title should represent better the idea and the results of the study.

·      Perhaps „emotional alterations“ in the title is too ambitious

·      Meridianins should be briefly explained in the Abstract

·      GluR1 was indicated as a keyword – it should be mentioned in the Abstract

·      Figure 1 should be removed into the Results. Besides, in Figure 1A sampling is indicated at 20, 60, and 90 minutes, whereas in the Figure legend 20 min, 1 h and 3 h are mentioned.

·      L50 - Akt is the major contributor to GSK3ß phosphorylation – I suggest to explain specifically effect of Akt phosphorylation at Ser9 on GSK3β activity

·    Welch’s correction is indicated only in 4.8. Statistics (as a part of Materials and Methods). It should be mentioned in the main text.

·      L25 - in some of these brain regions – it would be better to specify regions

·      L47 – ….it (GSK3β) is inhibited in response to stimulation, but showed increased activity or expression of GSK3β in mice subjected to different stress protocols (L53) – confusing

·      Ref. 19, 54 are not complete (journal, year, volume, pages)

·      L156 - excitatory glutamate receptor GluR1 – usually AMPA is named as a receptor, GluR1 is one of AMPA receptor subunits

·      Paragraph considering limitations of meridianins in the MDD therapy should be added (relatively high toxicity, no effect in forced swim test which is the best indicator of depression-like behaviour, …)

·      English needs to be improved (here are some suggestions)

-        L21 - excessive off-target effects and undesired secondary sequels situated GSK3β inhibitors in and impasse

-        L62 - First, the therapeutic effect of lithium is lower than the IC50 of lithium for inhibition of GSK3 – unclear, therapeutic effect is compared with the IC50 (number)

-        L66 placing the research seeking for modulators of the kinase in an impasse

-        L121 - These results confirm that intraventricular injection of and inhibits GSK3β – words are missing

-        L140 - redout

-        L282 – GluR1

-        L72 - [27– 30][31]. -please check

Author Response

Dear reviewer

We would like to re-submit our manuscript entitled "Meridianins Inhibit GSK3β In Vivo and Improve Behavioural Alterations Induced by Chronic Stress" by Sancho-Balsells et al. Our work is original research, it has not been previously published and it has not been submitted for publication elsewhere while under consideration. All authors declare there is no conflict of interest.

We thank the reviewers and the editors for their thoughtful comments. We have addressed all the questions that required new experimentation and changes in text sections. We honestly think that such changes strengthened the paper, and we indicate its exact new location in our point-by-point reply.

Please note that the modified text in the main text appears into the manuscript marked up in red.

REVIEWER 3

The authors investigated effects of meridianins (marine molecules from the Antarctic marine organisms) on the activity of GSK3β and related signalling pathways as GSK3β inhibitors are considered a promising therapeutic option in major depression disorder (MDD). They presented that meridianins act as a GSK3β inhibitors and modulators of PKA, PKC and Akt signalling in a brain region-specific manner. They also found that meridianins increased synaptic activity, specifically in the CA1 area of the hippocampus, probably due to the increased activity at the GluR1 subunit of the glutamate receptors. Finally, they showed that meridianins improve some chronic unpredictable mild stress (CUMS)-induced parameters such as weight loss, increased anxiety and cognitive impairment. The study is relevant and interesting, and the manuscript is easy to follow.

However, in my opinion, some additional experiments are needed before considering acceptance.

  • Meridianins are a very wide term including many potential active compounds. It is highly recommended to identify more precisely the active compound(s) from the extract.  Some meridianins (C,D,F,G) have been successfully synthesized (https://www.mdpi.com/1420-3049/27/7/2233/htm;).

We agree with the reviewer (as also pointed out by reviewer #2) that this information was missing in the manuscript. The extract used in our assays was a mixture because there was not enough amount of material to isolate all the different compounds in the mixture. Although there are 8 main meridianins (A to H; https://marinlit-rsc-org.sire.ub.edu/compounds?s=meridianin checked on August 16th, 2022), there are many other related meridianins and dimers in the extracts of these animals in minor amounts [5] and It is a very challenging task to purify them from the mixture. That’s the reason why we have used mixtures also in previous studies [6]. In order to expose these weaknesses, we have added a “limitation paragraph” in the discussion stating (page 10, lines 314-324) that:

“Finally, this study presents some limitations. First, future work should clarify whether all the meridianins in the mixture or only some of them, or perhaps some of them acting synergistically, are the direct responsible molecules for the observed effects. These could be done by using synthesized molecules, although not all the meridianins have been synthesized so far. Synthesis has been achieved for some meridianins and analogs through different pathways [80–87]. Moreover, although we demonstrated a GSK3β inhibition by meridianins, we cannot exclude the possibility of alternative mechanisms of action that could be mediating the improvements observed in stressed mice when treated with meridianins. Future studies should be focused on elucidating the targets altered by stress that are modulated by meridianins.”

  • It is already known that meridianins are potent kinase inhibitors, including the GSK-3 kinase (https://pubmed.ncbi.nlm.nih.gov/22512550/; https://www.mdpi.com/1660-3397/15/12/366/htm: https://pubmed.ncbi.nlm.nih.gov/32326204/;), and the authors have already shown that they are capable to improve cognitive deficits (https://www.ncbi.nlm.nih.gov/pmc/articles/PMC8908099/). In that way, many results presented here do not bring progress in the research field. The novelty here is related to meridianins and the MDD. Although the authors mentioned that „Future studies should be focused on elucidating the targets altered by stress that are modulated by meridianins ”, it is obligatory to study the effects of meridianins on GSK3β, PKA, PKC, Akt, CREB and GluR1 in the brain regions(prefrontal cortex, hippocampus, amygdala, nucleus accumbens) of the CUMS animals (and compare them with the effects observed in the CNT VEH mice) to confirm that the beneficial effects of meridianins in animal model od MDD are indeed GSK3β-related.

We agree with the reviewer that these studies are mandatory. Indeed, all reviewers have noticed that, and we have performed new experiments accordingly. In the following text you can see the homogenized response to all three reviewers:

We agree with the reviewer that this information was missing. To accomplish with the reviewer request we initially analyzed the effect of stress and meridianins on GSK3β. To do so, we performed Western Blot analysis. We first found that those stressed mice (CUMS VEH) presented an hyperactivation of GSK3β (See Figure 1B and 1C). This result goes in line with the literature and support our hypothesis. We then studied the effect of meridianins in those stressed mice. Contrary to what was expected, we could not find any difference between those CUMS mice treated with VEH or treated with MER (See attached figure). We then analyzed signaling pathways downstream GSK3β and we could not find differences between any group (See attached figure). We concluded that extensive manipulation (Invasive surgeries + long-term behavioral experimentation + the presence of an intra-cerebral mini-pump for four weeks) have altered, probably, the phosphorylation levels of GSK3B and its downstream signaling in a uncontrollable fashion. Indeed, in our team we have widely observed that mice with extensive manipulation have severe alterations in many molecular parameters that in basal states are changed. One example from our own lab in this sense has been recently published

 Fernández-García S, Conde-Berriozabal S, García-García E, Gort-Paniello C, Bernal-Casas D, García-Díaz Barriga G, López-Gil J, Muñoz-Moreno E, Soria G, Campa L, Artigas F, Rodríguez MJ, Alberch J, Masana M. M2 cortex-dorsolateral striatum stimulation reverses motor symptoms and synaptic deficits in Huntington's disease. Elife. 2020 Oct 5;9:e57017. doi: 10.7554/eLife.57017.

In this previous publication [1] we have observed that in a muse model of Huntington’s disease, upon massive manipulation, it losses its main biochemical hallmarks such as altered levels of DARPP32, BDNF, NMDAR, etc.

Rebuttal figure 1 (See attachment). These results are only depicted in the rebuttal letter for reviewer commodity. As stated below, we do not think that they help to improve the manuscript and that is why we put them only here.

Also, we performed the sampling 24h after the last day of treatment (minipumps last for 28 days and we performed brain sampling at day 29 post mini-pump implantation). Looking at the acute effects depicted in figure 1 regarding to GSK3β inhibition (from 20 to 60 minutes), it is conceivable that this timing could also exert some effect on this apparent lack of GSK3β inhibition and the subsequent changes on its downstream signaling.

Thus, we have not included this negative result because could be confusing and it is not providing useful information to the manuscript and, furthermore, we already showed modulation of GSK3β in vitro [2] and in vivo (Figure 1 of the present manuscript).

Finally, another possibility is that meridianins could also act through other mechanisms. In this sense, in the current version of the manuscript we have included a “weaknesses paragraph” at the end of the discussion section and we have stated that “although we demonstrated a GSK3β inhibition by meridianins, we cannot exclude the possibility of alternative mechanisms of action that could be mediating the improvements observed in stressed mice when treated with meridianins. Future studies should be focused on elucidating the targets altered by stress that are modulated by meridianins” (page 10, lines 314-324).

We sincerely thank to the reviewer for this important and useful observation.

  • The study is performed with a single concentration of meridianins (500 nM). Usually, more than one dose is applied to monitor the dose-dependency of the activity.

The reviewer is right and we have tried before different doses for other assays, and therefore for this study we used the one that was best based on our previous studies [6].

  • Figure 1 (in vivo inhibition of GSK3β) presents results in healthy animals exposed to meridianins for a short period of time (20 min – 3 h). In general, it is visible that activity of GSK3β returns to control values after 3h in all brain areas. How is this acute effect relevant to the long-term therapeutic interventions that are assumed for MDD treatment? Why chronic treatment was not performed to see (if any) long-term effects of meridianins?

As the reviewer already pointed out, since GSK3β inhibition by meridianins is transient, we chose the minipump treatment in order to perform a continuous delivery and the get a permanent GSK3β inhibition.

This is relevant from a point of view of intervention during a period of stress that could cause with high probability MDD-related symptoms. The relevance could also be from a preventive point of view.

Regarding to the long-term effects. As already pointed out in previous points from other reviewers, it is noteworthy that GSK3β inhibition is acute (0-60 minutes after treatment). Second, new experiments required by the reviewers #1 and #2 showed that changes by meridianins on GSK3β inhibition as well as on its downstream signaling pathways are not detected in the behavioral experiment because, probably, these brain samples were collected 24h after the last day of minipump treatment and from figure 1 we know that GSK3β inhibition lasts for 60 minutes. HOWEVER, alternative pathways modulated by meridianins, apart of GSK3β inhibition, could also exert long-term effects. Nevertheless, to test this alternative was beyond the scope of the present study.

THEREFORE, we have added a weaknesses section in the discussion contemplating, at least in part, these questions (page 10, lines 314-324):

“Finally, this study presents some limitations. First, future work should clarify whether all the meridianins in the mixture or only some of them, or perhaps some of them acting synergistically, are the direct responsible molecules for the observed effects. These could be done by using synthesized molecules, although not all the meridianins have been synthesized so far. Synthesis has been achieved for some meridianins and analogs through different pathways [80–87]. Moreover, although we demonstrated a GSK3β inhibition by meridianins, we cannot exclude the possibility of alternative mechanisms of action that could be mediating the improvements observed in stressed mice when treated with meridianins. Future studies should be focused on elucidating the targets altered by stress that are modulated by meridianins”

Other comments

  • Chronic stress is used to model MDD. Throughout the manuscript, the authors are focused on MDD and meridianins as the potential therapeutics. However, in the title chronic stress is emphasized which is confusing. The title should represent better the idea and the results of the study.

We agree with the reviewer that the manuscript was too focused on MDD rather than chronic stress. We have changed the text to center the attention on chronic stress and mood disorders rather than only MDD.

Some examples of the changes made are the following ones:

Line 46: “associated with MDD is the glycogen..” to “associated with stress-related pathologies is the glycogen..”

Line 239: “we focused on brain regions that are known to play important roles in MDD” to “we focused on brain regions that are known to play important roles in chronic stress”

Line 269-270: “In this line, alterations in the levels and function of GluR1 have been already described in MDD patients” to  “In this line, alterations in the levels and function of GluR1 have been already described in patients with mood disorders”.

  • Perhaps „emotional alterations“ in the title is too ambitious

We agree with the reviewer that the title was too ambitious. We have now changed the title to better suit the main results obtained in the manuscript. The new title proposed is: “Meridianins Inhibit GSK3β In Vivo and Improve Behavioural Alterations Induced by Chronic Stress”

  • Meridianins should be briefly explained in the Abstract

We agree with the reviewer that more information about meridianins was necessary in the Asbtract.

“Meridianins are alkaloids with an indole framework linked to an aminopyrimidine ring from Antarctic marine ascidians.”

See line 21-23

  • GluR1 was indicated as a keyword – it should be mentioned in the Abstract

We agree with the reviewer. We have modified the abstract in accordance with the results obtained.

  • Figure 1 should be removed into the Results. Besides, in Figure 1A sampling is indicated at 20, 60, and 90 minutes, whereas in the Figure legend 20 min, 1 h and 3 h are mentioned.

We apologize for the mistake. We have corrected the sampling time points in the Figure 1A.

  • L50 - Akt is the major contributor to GSK3ß phosphorylation – I suggest to explain specifically effect of Akt phosphorylation at Ser9 on GSK3β activity

We thank the reviewer for the suggestion. We have added this information in the introduction.

“Akt is the major regulator of GSK3β as it exerts an inhibitory phosphorylation on Ser9 in the amino-terminal part of the protein

Line 50-53

  • Welch’s correction is indicated only in 4.8. Statistics (as a part of Materials and Methods). It should be mentioned in the main text.

We apologize for this mistake, we never used the Welch’s correction. We have removed this information from the manuscript.

  • L25 - in some of these brain regions– it would be better to specify regions

We have now added this information in the abstract.

  • L47 – ….it (GSK3β) is inhibited in response to stimulation, but showed increased activity or expression of GSK3β in mice subjected to different stress protocols(L53) – confusing

We agree with the reviewer that the information was not clear enough. We have modified this part of the introduction.

  • Ref. 19, 54 are not complete (journal, year, volume, pages)

We apologize for the mistake. We have now corrected all the citations.

  • L156 - excitatory glutamate receptor GluR1– usually AMPA is named as a receptor, GluR1 is one of AMPA receptor subunits

We thank the reviewer for his appreciation. We have modified the text accordingly.

“Moreover, we also found that meridianins increased the activity of the excitatory glutamate receptor subunit GluR1 in the hippocampus”

 Paragraph considering limitations of meridianins in the MDD therapyshould be added (relatively high toxicity, no effect in forced swim test which is the best indicator of depression-like behaviour, …)

We agree with the reviewer that this paragraph was necessary. We have now added this information at the end of the discussion section. See page 9, lines 314-322.

  • English needs to be improved (here are some suggestions)

We have done our best to improve the English in our manuscript. We are also very thankful to the reviewer for all the suggestions.

-        L21 - excessive off-target effects and undesired secondary sequels situated GSK3β inhibitors in and impasse

We have now corrected this mistake in the text.

-        L62 - First, the therapeutic effect of lithium is lower than the IC50 of lithium for inhibition of GSK3 – unclear, therapeutic effect is compared with the IC50 (number)

We apologize for the mistake. We have corrected the text.

“First, the therapeutic dose of lithium is lower than the IC50 of lithium for inhibition of GSK3 [7].”

-        L66 placing the research seeking for modulators of the kinase in an impasse

We agree with the reviewer that the sentence need further clarification. We have corrected now in the text.

“Third, the use of other GSK3β inhibitors has been associated with excessive undesired and secondary effects [8,9] placing the research for kinase modulators in an impasse.”

-        L121 - These results confirm that intraventricular injection of and inhibits GSK3β – words are missing

We have now corrected this mistake in the text.

-        L140 – redout

We have now corrected this mistake in the text.

-        L282 – GluR1

We apologize for this point because we are not sure about its meaning. If it is about the use of GluR1 instead of GluA1, we observed that both are accepted and, indeed, recent papers published in 2022 are still using the nomenclature GluR1.

-        L72 - [27– 30][31]. -please check

We have corrected this mistake with the reference.

REFERENCES

  1. Fernández-García, S.; Conde-Berriozabal, S.; García-García, E.; Gort-Paniello, C.; Bernal-Casas, D.; Barriga, G.G.D.; López-Gil, J.; Muñoz-Moreno, E.; Soria, G.; Campa, L.; et al. M2 Cortex-Dorsolateral Striatum Stimulation Reverses Motor Symptoms and Synaptic Deficits in Huntington’s Disease. Elife 2020, 9, 1–24, doi:10.7554/ELIFE.57017.
  2. Llorach-Pares, L.; Rodriguez-Urgelles, E.; Nonell-Canals, A.; Alberch, J.; Avila, C.; Sanchez-Martinez, M.; Giralt, A. Meridianins and Lignarenone B as Potential GSK3β Inhibitors and Inductors of Structural Neuronal Plasticity. Biomolecules 2020, 10, doi:10.3390/BIOM10040639.
  3. Kuti, D.; Winkler, Z.; Horváth, K.; Juhász, B.; Szilvásy-Szabó, A.; Fekete, C.; Ferenczi, S.; Kovács, K.J. The Metabolic Stress Response: Adaptation to Acute-, Repeated- and Chronic Challenges in Mice. iScience 2022, 25, doi:10.1016/j.isci.2022.104693.
  4. Dang, R.; Wang, M.; Li, X.; Wang, H.; Liu, L.; Wu, Q.; Zhao, J.; Ji, P.; Zhong, L.; Licinio, J.; et al. Edaravone Ameliorates Depressive and Anxiety-like Behaviors via Sirt1/Nrf2/HO-1/Gpx4 Pathway. J. Neuroinflammation 2022, 19, 1–29, doi:10.1186/s12974-022-02400-6.
  5. Núñez-Pons, L.; Nieto, R.M.; Avila, C.; Jiménez, C.; Rodríguez, J. Mass Spectrometry Detection of Minor New Meridianins from the Antarctic Colonial Ascidians Aplidium Falklandicum and Aplidium Meridianum. J. Mass Spectrom. 2015, 50, 103–111, doi:10.1002/JMS.3502.
  6. Rodríguez-Urgellés, E.; Sancho-Balsells, A.; Chen, W.; López-Molina, L.; Ballasch, I.; Castillo, I. del; Avila, C.; Alberch, J.; Giralt, A. Meridianins Rescue Cognitive Deficits, Spine Density and Neuroinflammation in the 5xFAD Model of Alzheimer’s Disease. Front. Pharmacol. 2022, 13, doi:10.3389/FPHAR.2022.791666.
  7. Jope, R.; Roh, M.-S. Glycogen Synthase Kinase-3 (GSK3) in Psychiatric Diseases and Therapeutic Interventions. Curr. Drug Targets 2012, 7, 1421–1434, doi:10.2174/1389450110607011421.
  8. Del Ser, T.; Steinwachs, K.C.; Gertz, H.J.; Andrés, M. V.; Gómez-Carrillo, B.; Medina, M.; Vericat, J.A.; Redondo, P.; Fleet, D.; León, T. Treatment of Alzheimer’s Disease with the GSK-3 Inhibitor Tideglusib: A Pilot Study. J. Alzheimer’s Dis. 2013, 33, 205–215, doi:10.3233/JAD-2012-120805.
  9. Bhat, R. V.; Andersson, U.; Andersson, S.; Knerr, L.; Bauer, U.; Sundgren-Andersson, A.K. The Conundrum of GSK3 Inhibitors: Is It the Dawn of a New Beginning? J. Alzheimer’s Dis. 2018, 64, S547–S554, doi:10.3233/JAD-179934.

Round 2

Reviewer 1 Report

Accept

Author Response

Dear reviewer

We would like to re-submit our manuscript entitled "Meridianins Inhibit GSK3β In Vivo and Improve Behavioural Alterations Induced by Chronic Stress" by Sancho-Balsells et al. Our work is original research, it has not been previously published and it has not been submitted for publication elsewhere while under consideration. All authors declare there is no conflict of interest.

We thank the reviewers and the editors for their thoughtful comments. We have addressed a last minor comment from the academic editor

Please note that the modified text in the main text appears into the manuscript by using the track changes option in the main word file.

ACADEMIC EDITOR

Authors must make the revisions requested by the reviewer 2. In the materials and methods in the "marine organisms" section the authors indicate that they tested a mixture of meridianins. In this sense, the authors must indicate which meridianins are present in the sample and from which source these meridianins were isolated. 

We agree with the academic editor, and we have performed the required changes in page 10, lines 340 and in lines 342-344. Here we put the text for the editor’s and reviewer’s commodity:

“Marine compounds were obtained from the available sample collections at the Uni-versity of Barcelona (BEECA Department, Faculty of Biology) from previous Antarctic pro-jects (BLUEBIO, CHALLENGE). Briefly, the collected Antarctic marine organisms of the species Aplidium falklandicum were extracted with organic solvents and the extracts were further purified through chromatographic methods (HPLC) as previously reported [36,37]. Samples were kept frozen at -20°C until used. In our assays, we used the mixture of sever-al meridianins (A-G) since the total sample amount was low and it was not possible to identify which meridianins were present in the mixture nor in which proportions, since this is a quite complex group of compounds [35]”

Reviewer 2 Report

I understood the authors' responses for the reviewers' comments.

Author Response

(The authors gave the same response as above.)

Reviewer 3 Report

English still could be improved.

Author Response

(The authors gave the same response as above.)
